# Characterisation of Particle Size and Viability of SARS-CoV-2 Aerosols from a Range of Nebuliser Types Using a Novel Sampling Technique

**DOI:** 10.3390/v14030639

**Published:** 2022-03-19

**Authors:** Susan Paton, Simon Clark, Antony Spencer, Isobel Garratt, Ikshitaa Dinesh, Katy-Anne Thompson, Allan Bennett, Thomas Pottage

**Affiliations:** UK Health Security Agency (UKHSA), Porton Down, Salisbury SP4 0JG, UK; simon.clark@phe.gov.uk (S.C.); antony.spencer@phe.gov.uk (A.S.); ig415@bath.ac.uk (I.G.); ikshitaa.dinesh@postgrad.manchester.ac.uk (I.D.); kt_anne82@yahoo.co.uk (K.-A.T.); allan.bennett@phe.gov.uk (A.B.); thomas.pottage@phe.gov.uk (T.P.)

**Keywords:** SARS-CoV-2, aerosols, particles, viable, respirable, sizing, sampling

## Abstract

Little is understood about the impact of nebulisation on the viability of SARS-CoV-2. In this study, a range of nebulisers with differing methods of aerosol generation were evaluated to determine SARS-CoV-2 viability following aerosolization. The aerosol particle size distribution was assessed using an aerosol particle sizer (APS) and SARS-CoV-2 viability was determined after collection into liquid media using All-Glass Impingers (AGI). Viable particles of SARS-CoV-2 were further characterised using the Collison 6-jet nebuliser in conjunction with novel sample techniques in an Andersen size-fractioning sampler to predict lung deposition profiles. Results demonstrate that all the tested nebulisers can generate stable, polydisperse aerosols (Geometric standard deviation (GSD) circa 1.8) in the respirable range (1.2 to 2.2 µm). Viable fractions (VF, units PFU/particle, the virus viability as a function of total particles produced) were circa 5 × 10^−3^. VF and spray factors were not significantly affected by relative humidity, within this system where aerosols were in the spray tube an extremely short time. The novel Andersen sample collection methods successfully captured viable virus particles across all sizes; with most particle sizes below 3.3 µm. Particle sizes, in MMAD (Mass Median Aerodynamic Diameters), were calculated from linear regression of log_10_-log_10_ transformed cumulative PFU data, and calculated MMADs accorded well with APS measurements and did not differ across collection method types. These data will be vital in informing animal aerosol challenge models, and infection prevention and control policies.

## 1. Introduction

Severe acute respiratory syndrome coronavirus 2 (SARS-CoV-2), the causative agent of coronavirus disease of 2019 (COVID-19) has been a causative factor in the deaths of more than of 5.6 million people worldwide [1]. There is currently no firm consensus on its routes of transmission, as evidence exists that pathogenic SARS-CoV-2 aerosol particles across a distribution of sizes, play a major role, and that both droplets and aerosols are implicated [2,3,4,5,6,7,8]. “True” aerosol transmission is considered to occur from droplet nuclei, particles < 5 µm, which can remain suspended in the air indefinitely and can penetrate the lower lung [9]. Particles between 5 and 10 µm can travel over shorter distances (metres, depending on surrounding air currents), and be inhaled into the upper respiratory tract; whilst even larger particles from 10 to 100 µm, most likely play a role in either direct transmission [10], where droplets land directly onto mucosal membranes from the respiratory secretions of another person; or in indirect (fomite) transmission, where they settle onto surfaces subsequently touched, and contamination is transmitted to the mucosal membranes of the face, causing infection [11]. Previous work investigated the survival of SARS-CoV-2 on different surfaces [12,13]. The work reported here focuses on the generation and size characterisation of SARS-CoV-2 in aerosol particles.

Further understanding of the viability and particle deposition profiles of SARS-CoV-2 will add knowledge to the transmission potential of the virus. In addition, these data will aid in the development of more reliable animal models to recapitulate COVID-19 disease in humans; to aid understanding of disease diagnosis, pathology, and to evaluate new therapeutics. Thus far, the majority of SARS-CoV-2 animal studies have employed inoculation of mucous membranes to cause disease: intranasally, intratracheally, orally, intraocularly, or a combination of routes [14,15,16,17,18]. Whilst these studies have resulted in successful infection, they reflect only direct and indirect contact transmission, and not aerosol-acquired infection. Infection route has been shown to influence disease severity for SARS-CoV-2 [19,20], as well as other respiratory diseases [21,22]. Thus, methods of producing stable aerosols of virus with known particle size distribution is key to the development of more informative animal model of SARS-CoV-2 infection by the aerosol route of transmission.

Our studies investigated the viability and size distribution of particles created by a number of different nebuliser types; 3- and 6-jet Collison nebulisers; two medical nebulisers (normally used to deliver therapeutics) [23,24,25,26,27]; and two sparging liquid aerosol generators (1 inch and 90 mm SLAGs). In terms of nebuliser function, the “gold standard” Collison, created in 1973 [28], generates aerosols by applying an airflow (at around 26psi) to a liquid suspension of microorganisms; creating a vacuum which draws the suspension up a tube into which are engineered jets (between 1 and 24, depending on model type). The liquid suspension exits the jets at high force, creating particles which impact onto the surrounding glass jar and break up into smaller particles, for delivery into the test system. This method is designed to deliver near mono-disperse aerosols, in the size range of 1 to 3 µm [29]; but is considered “harsh” on microorganisms, applying both shearing and impaction forces which can render a proportion of the organisms non-viable. It also re-circulates the liquid suspension meaning that over longer spray times, the viability of the suspension can decrease. As SARS-CoV-2 is an enveloped virus, it is thought to be more prone to damage from the environment than unenveloped viruses. We therefore investigated nebulisers employing different aerosolization methods for comparison.

Of the medical nebulisers employed, the Omron MicroAIR U22 contains a finely perforated mesh, across which an electric current is applied. The current causes vibration of the mesh at 180 kHz, which aerosolises the liquid suspension sitting atop it, to produce fine particles for delivery to the lower respiratory tract [30]. The Pari LC Sprint Star utilises the Venturi principle, where compressed air draws liquids through a narrow orifice to impact on the inside of a tube. This impaction creates particles of varying sizes; the larger particles are removed by baffles, to generate a fine particle aerosol [31].

Sparging Liquid Aerosol Generators (SLAGs), are considered to be one of the “gentlest” forms of nebulisers, as they are thought to mimic natural aerosol generation in the respiratory tract more closely through the bursting of bubbles [32]. SLAGs function at low pressure by applying air to a perforated disc, wetted by the microbial suspension, creating bubbles; bursting of the bubbles generates the desired aerosols, with no shearing or impaction forces, nor any recirculation of the liquid. The larger disc diameter version allows for larger volumes of aerosol delivery [33].

An additional part of the study investigated whether an Andersen size-fractionating impaction sampler using novel virus collection methods, cell culture medium and gelatine filters, could efficiently measure the particle size distribution of aerosols of SARS-CoV-2 generated by a 6-jet Collison nebuliser.

The aims of these studies were to determine the survival of SARS-CoV-2 in aerosols generated by these nebulisers and to characterise their sizes; to allow the assessment of the risk of transmission, and generate data for future in vivo studies that use aerosol infection as the route of delivery.

## 2. Materials and Methods

### 2.1. SARS-CoV-2 Stock

The SARS-CoV-2 isolate, England 02/2020 (EPI_ISL_407073) passage 3 (P3) used in this study was propagated by the High Containment Microbiology Department at UKHSA (formerly PHE), Porton Down, Salisbury, UK. The virus was isolated from a clinical sample taken during acute phase illness, using Vero E6 cells (ECACC, 85020206). A P2 master bank was produced using Vero E6 cells (BEI Resources, NR-596) and a P3 working bank produced in Vero/hSLAM cells (ECACC 04091501). Cell lines were infected at 95% confluence with an of MOI 0.0005–0.01 and maintained in 1× Minimal Essential Medium GlutaMax (Gibco, Grand Island, New York, USA), 4% heat-treated foetal bovine serum(Sigma, St Louis, MO, USA), 1× non-essential amino acids (Gibco, Paisley, Scotland, UK), 25 mM HEPES buffer (Gibco, Paisley, Scotland, UK); additionally, Vero/hSLAM cells were also maintained in the presence of 0.4 mg/mL geneticin (Gibco, Grand Island, New York, USA ). Virus was harvested 3 to 6 days post-infection and supernatant clarified by centrifugation (3000 rpm, 10 min). Virus was aliquoted and stored at −80 °C. The titre of the P3 virus stock was determined to be 2.0 × 10^7^ PFU (plaque forming units)/mL by plaque assay. All work handling SARS-CoV-2 was performed within a containment level 3 laboratory.

### 2.2. Nebulisers

Six different nebulisers, with four types of nebulising action, were selected to aerosolise SARS-CoV-2 under different relative humidities (RH). The nebulisers are listed in Table 1. in an order relevant to the degree of physical impact of the aerosolization process on the pathogen (shear forces generated from pressure, from highest to lowest):

SLAG operation: To avoid damaging the perforated disc elements, the pressure was gradually increased by increasing the air flow applied in increments for each run. Similarly, the volume of virus inoculum applied during operation was altered to produce visible, vigorous bursting bubbles without excessive foaming, by increasing and decreasing the speed of the peristaltic pump used to aliquot the liquid. It was found, especially for the 90 mm SLAG, that pre-wetting the perforated 90 mm disc moments before applying compressed air to the system, achieved maximal levels of bubble bursting (sparging). Due to limitations on time, the air pressure and inoculum volume were not fully optimised for the 1 inch SLAG, however the ranges applied to both disc sizes appear in Table 1. All other nebulisers were used according to the manufacturer’s instructions.

### 2.3. Henderson Apparatus

A Henderson apparatus allows closed-circuit containment and direction of microbial aerosols generated from a nebuliser at a controlled relative humidity (Figure 1). The Henderson apparatus was used to deliver aerosols of SARS-CoV-2 at 40 L/min (balancing the total flow between the nebuliser and the Biaera unit) and the aerosols were collected at the end of the apparatus spray tube using biological samplers. The apparatus was contained within a flexible film isolator (FFI) within Containment Level 3 facilities. The challenge system is controlled by an AeroMP control unit (Biaera, Hagerstown, MD, USA). The AeroMP is a platform system designed to manage the aerosol generation, characterisation and sampling processes via a dashboard software laptop system. The aerosol management platform controls, monitors, and records all relevant parameters during an aerosol procedure including air flow rates, generated pressures, temperature and relative humidity. The software automatically monitors the conditions in the apparatus and balances the system airflows. The mean temperature during all experiments was 23.4 °C, range 21 °C to 25 °C.

### 2.4. Aerodynamic Particle Sizing Spectrophotometer

An APS device (model 3321, TSI Inc., Shoreview, MN, USA) was used to determine the size distributions of particles in aerosolised viruses, nebulised, and routed through the Henderson Apparatus. The APS device was set to sample 1:100 dilution, for two 30 s periods at 30 s and 4 min post-initiation of nebulisation. The sample was taken at the end of the spray tube and in an equivalent location to where animals would theoretically be exposed to the aerosol. See Figure 1.

### 2.5. Biological Samplers

To assess viability of nebulised viruses, all-glass impingers (AGI-30) containing 10 mls of cMEM (complete Minimal Essential Medium [12]) and operating at 12 L/min were connected to the Henderson apparatus down-stream of the spray tube (see Figure 1) for the duration of 5-min sprays [34].

A 6-stage Andersen sampler was used to fractionate the aerosols generated by the 6-jet Collison nebuliser into different sizes during sampling [35]. Relative humidity was maintained within a range of 45 to 60%. The Andersen was operated for 5 min at 28.3 L/min and was placed at the same point as the AGI-30 in the previous experiments, see Figure 1. Three different collection methods were employed in the Andersen’s stages (differing from the normal solid agar media used) to ensure the viral particles could be captured and enumerated, using glass petri dishes to reduce the effects of static charge: (a) 27 mL of cMEM (stages 1–5 only); (b) 20 mL 2% agarose plus 7 mL cMEM (6 stages); or (c) Gelatine Membranes (6 stages), 0.2 µm pore size (Sartorius, Goettingen, Germany). Four sterilised glass microscope slides were stacked under each gelatine membrane, to raise them to the correct height for size-based impaction to occur. Post-exposure, gelatine membranes were dissolved in 10 mL warmed cMEM for 1 min with agitation, before collecting for assay.

All biological sampling liquids described above were transferred to a cryotube for storage at −80 °C before analysis. The process of storing samples at −80 °C then thawing before processing was not found to significantly affect viability of the virus [12].

### 2.6. Plaque Assays

Plaque assays were completed according to protocols described previously [12]. Briefly, the thawed collection fluid was serially diluted 1 in 10 to an appropriate dilution, then plaque assayed on Vero-E6 cells, before staining and enumeration of plaques. The number of plaques in each well was determined and expressed as PFU.

### 2.7. Data Analysis

Virus titre in samples collected from nebulisers, AGI and Andersen samplers was calculated by taking averages of technical replicates (plaque assay performed in duplicate), to give PFU/mL; this was then multiplied by the total liquid volume of the sample to give total PFU.

Viable fraction (VF) values were calculated for each nebuliser assessed. The VF is the relationship of viable virus particles arising from the total number of particles nebulised. It is found from the total PFU collected in the AGI, divided by the total number of particles counted by the APS during the 5-min spray. The units are PFU/particle.

Spray Factors (SF) were calculated for each nebuliser assessed. The SF is a unitless ratio that defines a relationship between the viability of a challenge suspension in the nebulizer (C_neb_, PFU/L) and the concentration of viable virus in the circulating aerosol (C_aero_, PFU/L) and is commonly used in animal aerosol challenge models [23,29,36]. The value obtained is system dependent, but it allows comparisons of stability between different agents used:$$\mathrm{SF}=\frac{C_{\mathrm{aero}}}{C_{\mathrm{neb}}}$$

The concentration of virus in the circulating aerosol (C_aero_) is calculated by the formula:$$C_{\mathrm{aero}}=\frac{C_{\mathrm{imp}}\times {\;V}_{\mathrm{imp}}}{Q_{\mathrm{imp}}\times {\;t}_{\exp}}$$
where C_imp_ is the concentration of virus (PFU/L), V_imp_ is the volume of collection fluid in the AGI (mL), Q_imp_ is the sample flow rate of the AGI (L/min), and t_exp_ is the exposure time (min). Virus concentrations in C_aero_ are converted to PFU/L to allow calculation against C_neb_ in PFU per litre of air sampled.

Calculation of Mass Median Aerodynamic Diameter (MMAD) size from Andersen data: Total PFU was used to calculate (in Microsoft Excel) cumulative percentage collected at each stage 1 to 6 as:$$Cumulative\;\%\;of\;stage\;\left(n\right)=\frac{\Sigma \left[total\;pfu\;of\;stages\;\left(n\to 6\right)\right]}{\Sigma \left[total\;pfu\;stages\;\left(1\to 6\right)\right]}\times 100$$

These percentages were plotted against the size cut-off for each stage, here the cut-offs for stages 1 to 6 are 7 µm, 4.7 µm, 3.3 µm, 2.1 µm, 1.1 µm, and 0.65 µm, respectively; and log_10_ transformed on both axes. A linear regression line was added and the particle size at 50% is then taken as the MMAD for that run. This took place in GraphPad Prism v9.

## 3. Results

### 3.1. Nebuliser APS Readings

All nebulisers produced particles in the respirable size range, as measured by the APS, Table 2. MMAD values tended to be smaller for each nebuliser at the lower RH range, but this was not statistically significant (*p* > 0.05). Across all RH values, aerosol particles with the lowest MMAD were produced from the PARI Sprint Star, compared to the Omron (*p* = 0.0012) and the 1 inch SLAG (*p* = 0.0132); no other MMAD results were significant. The PARI SprintStar also had the lowest level of variance in the size distribution, calculated by geometric standard deviations (GSD) (ca. 1.6); indicating production of mono-disperse aerosols of ca. 1.2 µm diameter from this nebuliser. The nebuliser which produced the widest distribution of particle sizes was the 1 inch SLAG (GSD of ca. 2); indicating high polydispersity. This was only significant compared to the GSDs of the Omron (*p* = 0.0018) and Sprint Star (*p* = 0.0134).

### 3.2. Nebuliser PFU Counts

Viable virus was consistently recoverable from the AGI and from all six nebuliser reservoirs following aerosolization for 5 min, across three different relative humidity ranges. The total PFU collected as an average (with standard deviation) across all RH values, were, in descending order: PARI SprintStar 2.76 × 10^5^ (s.d. 5.06 × 10^4^), SLAG 90 mm 2.26 × 10^5^ (s.d. 3.92 × 10^5^), Collison 6-jet 1.49 × 10^5^ (s.d. 1.41 × 10^5^), Collison 3-jet 1.21 × 10^5^ (s.d. 3.47 × 10^4^), Omron MicroAIR U22 1.00 × 10^5^ (s.d. 8.99 × 10^4^) and SLAG 1 inch 1.39 × 10^4^ (s.d. 2.16 × 10^4^). From these data, VF values of SARS-CoV-2 aerosols have been calculated, to estimate the impact on viability of physical forces experienced during nebulisation, with lower values indicating a larger impact on viability. Figure 2 shows that median VF values across all RH values were between 6.7 × 10^−6^ (Collison 6 jet) to 2.6 × 10^−2^ (90 mm SLAG). Overall, the SLAG nebulisers resulted in higher VF values (suggestive of their gentler mode of action on aerosol generation), compared to the four other nebuliser types, though only the 90 mm SLAG results were significantly higher than the 3-jet and 6-jet Collisons (*p* = 0.010). Humidity did not significantly impact VF values (*p* = 0.177), but values were slightly less variable at lower RH, with coefficients of variation for high, intermediate, and low RH of 206.3%, 235.2%, and 125.5%, respectively.

### 3.3. Spray Factors

Spray factor values were also calculated across all RH values; the PARI Sprint Star produced the highest SF, with the 3- and 6-jet Collisons and the Omron within 1 log_10_ of this; SF for the SLAGs were the lowest of all. For each nebuliser the median and standard error are given below (in descending order): PARI SprintStar 2.77 × 10^−6^ (1.83 × 10^−5^); Collison 6-jet 2.23 × 10^−6^ (1.97 × 10^−6^); Collison 3-jet 1.85 × 10^−6^ (8.70 × 10^−7^); Omron MicroAIR U22 1.66 × 10^−6^ (1.20 × 10^−6^); 90 mm SLAG 6.45 × 10^−7^ (4.31 × 10^−7^); 1 inch SLAG 3.19 × 10^−8^ (7.5 × 10^−8^). The SF generated from the PARI SprintStar was significantly higher than that from the 1 inch SLAG, (*p* = 0.012); but no other SF’s differed between nebulisers. Humidity did not significantly impact SF values (*p* = 0.609).

### 3.4. Andersen Sampling Efficiency

The novel virus collection methods employed on the Andersen sampler were comparable to each other in overall efficiency, and particle size fractionation capability, when SARS-CoV-2 was aerosolised from a Collison 6-jet nebuliser. Figure 3 shows the comparison in biological sampling efficiency for the three sampling methods employed. For all three, the majority of SARS-CoV-2 particles were collected on the lower stages of the sampler, <3.3 µm. The liquids in the AGI samplers, collected from the Collison 6-jet (first study), contained 4.48 × 10^3^ pfu/L, (standard deviation 2.39 × 10^3^), which is comparable to the PFU/L collected by the Andersen methods: total PFU recovered across all stages, per litre of air sampled (with standard deviations) were: 27mL cMEM, 3.8 × 10^3^ pfu/L (s.d. 2.7 × 10^3^); 7 mL cMEM, 7.1 × 10^2^ pfu/L (s.d. 6.9 × 10^2^); and Gelatine, 2.0 × 10^2^ pfu/L (s.d. 1.4 × 10^2^). The total PFU/L across the three Andersen methods and AGI sampler were not significantly different (*p* = 0.241).

### 3.5. Andersen MMAD Results

The PFU collected across each stage allowed the calculation of MMAD for each media type. These numbers correspond well to the MMADs as measured by the APS (Table 3). The GSD, as measured by the APS, denotes polydispersity of particles around 1.7 µm, from the 6-jet Collison, which corresponds to the earlier measurements at different RH ranges (Table 3). The collection methods are not statistically different from each other, in terms of MMAD.

## 4. Discussion

This study demonstrates that both established and novel nebulisers can be used to generate stable and viable SARS-CoV-2 aerosols. The 90 mm SLAG generated aerosols with the highest viability fraction; demonstrating a gentle nebulisation action. Aerosols generated at lower RH produced fewer variable data, but overall RH did not significantly influence the viability, size or numbers of viable particles produced. The Hendersen apparatus, where generated aerosols pass from the nebuliser to the point of delivery (AGI or animal challenge equipment) within 40 microseconds, is designed to ensure even mixing of the aerosol, rather than its conditioning. In order to observe any effects of humidity on aerosols, a system such as the Goldberg drum should be employed. In terms of particle sizing and size distributions, the PARI Sprint Star produced the particles of smallest MMAD and GSD, and was the producer of the highest number of viable particles overall. To this author’s knowledge, no work characterising microbial aerosols from the PARI Sprint Star has been produced, thus representing an exciting avenue for future research in experimental microbial aerosol generation.

In others’ work characterising nebulisers (including a Collison nebuliser and the Omron MicroAir U22), in creation of airborne virus aerosols, Niazi et al. [37] found that the Collison reduced viability of generated influenza aerosols more than the other nebulisers studied. However, the nebulisation time was six times longer than this study, and the re-circulating nature of the Collison means it is less suited to such long spray times. However, the GSDs measured from the Collison and Omron nebulisers in that work accord well with ours, suggesting a similar particle size distribution, even between different virus types. Others, working with the SLAG to generate viral aerosols have had less success; Fennelly et al. failed to detect either viral RNA nor viable flu particles from SLAG nebulisation [38]. The SLAG systems, in this author’s opinion, represent an interesting avenue for further research for fragile microbial aerosols, especially as this study demonstrated the potential in further optimisation of the SLAG 1 inch.

Further APS data produced during this study included the CMAD value (count median aerodynamic diameter), a size calculation based on the number distribution of an aerosol. Due to the large numbers of very small particles (less than 1 µm in diameter) produced by nebulisers, CMAD will give smaller size calculations compared to MMAD. However, CMAD is less used in the study of microbial aerosols, as, the smaller a particle is, the less likely it is to contain a virion/bacterium. Traditionally, cascade samplers are employed to measure the size distribution of the viable particles in microbial aerosols, through calculation of MMAD as described. Thus, our Andersen data and APS MMAD data are the most relevant in describing particle size distributions in this study.

This study was designed to determine the effectiveness of the nebulisers’ interactions with SARS-CoV-2, to understand and inform the design of effective animal models utilising an aerosol infection route. Determination of spray factor allows one to calculate the presented dose to an animal, based on any given concentration of a pathogen in a nebuliser and is specific both to the aerosol challenge system (apparatus, mode of aerosolization, etc.) and the individual organism. For example, another respiratory pathogen, Influenza H1N1, delivered via a 6-jet Collison in the same Biaera/Hendersen system as used here has a SF of 1.32 × 10^−6^ [39,40] which is comparable to our derived value for SARS-CoV-2 of 2.23 × 10^−6^ in the same system. Our study also showed that the SLAG, mesh and jet nebulisers were no more likely to inactivate SARS-CoV-2 than either of the more standard Collison nebulisers, within our system. The lack of significant influence of RH, or even nebuliser type (excluding the 1-inch SLAG) on spray factors, demonstrates that the practice of using Collison nebulisers is predicted to not have a negative impact on the viability of particles generated in a SARS-CoV-2 infection model. Specific spray factor values calculated in this study will be used to determine the starting inoculum in an aerosol challenge system, required to give a specific presented dose (or range of doses) to animals. These data will allow improved and reproducible infection and interpretation of disease development and assessment of interventions in this model.

This study also demonstrated three novel aerosol collection methods for use within an Andersen sampler, which, overall, had comparable collection efficiencies compared to the AGI sampler for SARS-CoV-2. This corroborates the work of others such as Kutter et al. [41], Fennelly et al. [42] and Kulkarni et al. [43] who also successfully collected respiratory viruses within an Andersen sampler used with viral collection methods such as agar, semi-solid gelatine, or liquid media. Our collection rates (in terms of PFU collected per litre of air sampled) were higher than those aforementioned studies, but small differences such as our use of glass petri dishes over plastic ones (known to influence efficiency due to static forces generated by air flows [35]), may have contributed to this difference.

This study was not designed to provide an in-depth functional analysis of different nebuliser types. Many others, employing tracers to characterise the physical efficiencies of nebulisers, have performed this thoroughly [24,25,26,27,28,44,45]. However, to this author’s knowledge, the efficacy of different nebuliser types has yet to be explored for SARS-CoV-2 [46]. Similarly, most animal studies exploring SARS-CoV-2 have challenged animals intra-nasally, intra-ocularly, or intra-orally, with few delivered by the aerosol route [14,15,16,17,18]. Only one recent study examined natural transmission between intra-nasally infected hamsters to naïve animals [47]. Our study sought to characterise the MMAD and viability of SARS-CoV-2 aerosols generated with different relative humidities from six nebulisers. To date, no other work has compared nebulisation effects on SARS-CoV-2 across such devices; nor attempted to characterise by size-fractionation, viable SARS-CoV-2 virus particles, using these methods.

The authors acknowledge the limitations of this study, which were largely influenced by access to key containment facilities during the COVID-19 pandemic. This resulted in a lack of time to optimise the parameters for the relatively novel SLAG nebulisers. These have not been often used in virus aerosolization studies, by this group or others, and reflects the investment needed to establish the most effective parameters. Facility access also limited the numbers of runs that could be completed across the ranges of RH explored. We acknowledge that the use of the large fill volume within the Andersen, and the need to exclude the lowest Anderson stage to prevent media overspill, makes it unwieldy and impractical for field studies. As recovery of viable virus from the 7 mL fill volume of the Andersen sampler compared to the AGI was only 0.81 log_10_ less, and particle size characterisation was accurate compared to the APS, we would recommend the 7 mL + agarose base collection method for use either in in vitro lab studies, or in the indoor built environment.

Future studies replicating this work will determine if any differences in viability or particle sizes arise between variant strains; which may help inform policy and provide important information for future aerosol studies in animals. Future work employing different aerosol capture methods, such as the Goldberg drum or spider microthreads, could explore decay rates over longer time periods, of virus in different temperature and relative humidities, and also incorporate electron microscopy images of the virus to determine physical effects of nebulisation compared to those observed in sputum. Future work may also involve comparison of natural transmission in animal models [47] to those animals infected via nebuliser-generated aerosols; such work may aid elucidation of the particle sizes and concentrations generated by intra-nasally infected animals, in aerosol infection models.

Detection of viable viruses such as SARS-CoV-2 in air remains difficult, due to their usually low concentration and the negative impacts on viability that air sampling forces have on fragile enveloped viruses. This work establishes a new method of detecting and fractionating such particles; and demonstrates that SARS-CoV-2 generated by nebulisers have the potential to transmit to the deep lung; highly relevant to in vivo models. These data will also inform further in vitro aerosol studies and set the foundation for in vivo studies designed to understand transmission and disease caused by the aerosol route of infection with SARS-CoV-2, and may aid in informing infection prevention and control policies for indoor air.

## Figures and Tables

**Figure 1 viruses-14-00639-f001:**
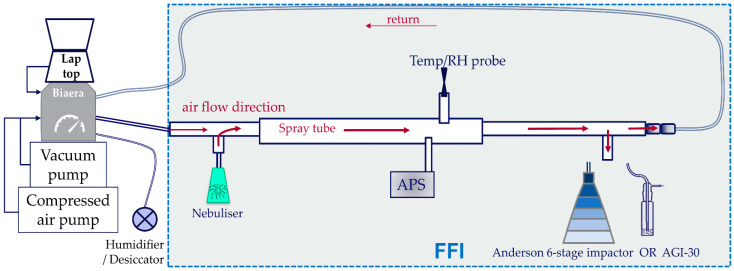
The Henderson apparatus and Biaera AeroMP unit set up at containment level 3. The shaded area represents the FFI (flexible film isolator). For Andersen validation, the Andersen sampler was placed at the same position as the AGI-30. Diagram is a representation of connections used.

**Figure 2 viruses-14-00639-f002:**
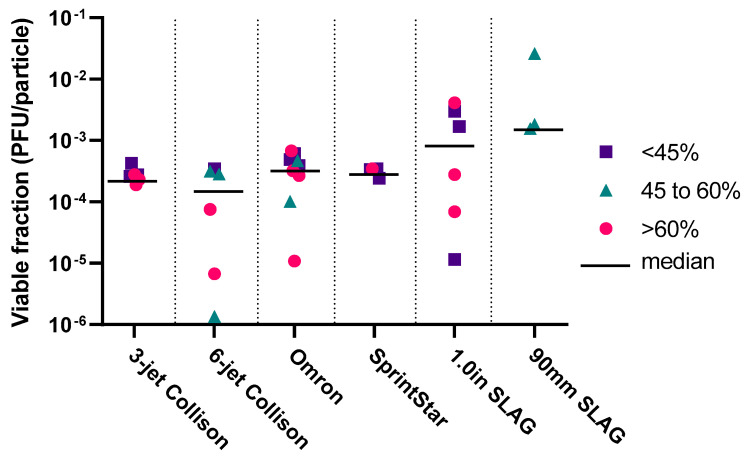
Viable fraction (VF) values for nebulisers at different relative humidity ranges. Each data point represents one biological repeat: magenta circles > 60% RH; teal triangles 45 to 60% RH; purple squares < 45% RH; black line is the median value across all RH’s for that nebuliser.

**Figure 3 viruses-14-00639-f003:**
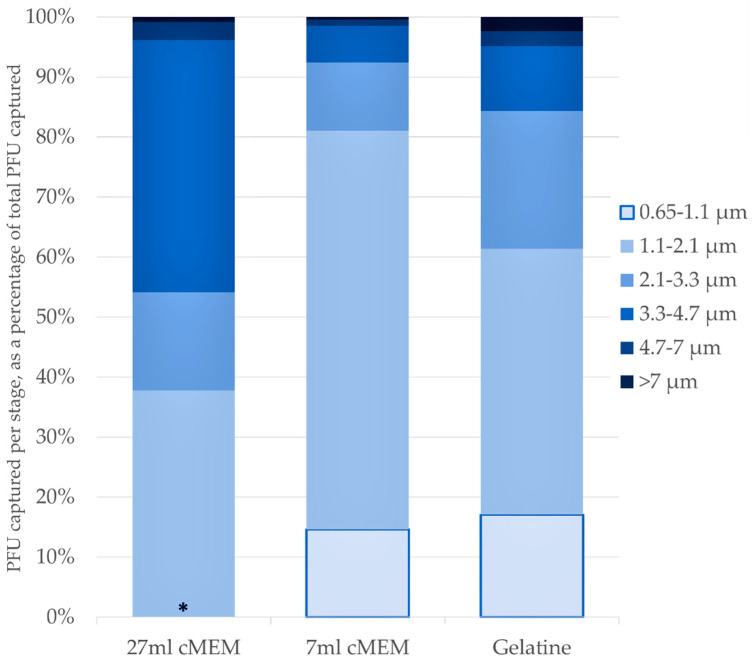
Proportion of total pfu captured by each Andersen stage, by sampling media type. ***** Stage 6 (0.65 µm to 1.1 µm) omitted from the 27 mL cMEM fill, due to overspill of liquid from the previous (1.1–2.1 µm) stage. *n* = 4 for 27 mL cMEM and 7ml cMEM, gelatine filter *n* = 3.

**Table 1 viruses-14-00639-t001:** Types and information on different nebulisers used in this study. Lpm = Litres of air per minute, psi = pounds per square inch.

Nebuliser	Manufacturer	Inoculum Vol.	Method of Aerosolisation	Air Flow Applied (lpm)	Pressure Generated (psi)
Collison 6-jet	CH technologies	10 mL	Impaction	17 ± 0.5	27.9 ± 1
Collison 3-jet				8.5 ± 1	27 ± 2
LC Sprint Star	PARI	8 mL	Jet	6	25 ± 1
Omron MicroAir U22	Omron	5 mL	Vibrating mesh	N/A	N/A
SLAG 1 inch	CH technologies	* Variable	Sparging liquid	* 6 to 14	3 to 7
SLAG 90 mm				* 6 to 30	0.6 to 5.2

* see note in text regarding SLAG operation.

**Table 2 viruses-14-00639-t002:** APS data for nebulised SARS-CoV-2 at varying relative humidities. MMAD, mass median aerodynamic diameter; GSD, geometric standard deviation, is a unitless number that denotes size distribution of the aerosols produced. GSD figures are presented as the range of values calculated by the APS for repeat runs.

RH Range	Result	Collison 3-Jet	Collison 6-Jet	Omron MicroAIR U22	PARI Sprint Star	SLAG 1 inch	SLAG 90 mm
>60%	MMAD ( µm)	1.57	1.60 ^	2.07 ^$^	1.25 *	2.19	ND
	GSD	1.85–1.88	1.85 ^	1.60–1.66 ^$^	1.66 *	1.91–2.06	ND
45–60%	MMAD ( µm)	ND	1.58	2.10 ^	ND	ND	1.34
	GSD	ND	1.85–1.92	1.66 ^	ND	ND	1.15–2.14
<45%	MMAD (µm)	1.38	1.60 *	1.81	1.20	1.77	ND
	GSD	1.78	1.92 *	1.60–1.78	1.60–1.66	1.84–2.16	ND

Runs were performed three times unless stated; * denotes *n* = 1, ^ denotes *n* = 2, ^$^ denotes *n* = 5. ND = not done.

**Table 3 viruses-14-00639-t003:** Particle sizes and size distributions of particles collected via three sampling methods in the Andersen, using both calculated and APS-measured values. Two values for MMADs (mass median aerodynamic diameters) were found by: (Left-hand side, MMAD (calc)); calculation from the cumulative percentage deposition of viable viral particles across the stages, and (Right-hand side), as measured by the APS. GSD average is found from the median values, as measured from the APS, and is a unitless number.

Andersen		APS	
Sample Type	MMAD (calc)	MMAD	GSD
27 mL	1.86 µm	1.59 µm	1.78 to 1.92
7 mL	1.49 µm	1.65 µm	1.84 to 1.92
GF	1.69 µm	1.74 µm	1.78 to 1.91
**Average**	**1.68 µm**	**1.66 µm**	**1.85**

## Data Availability

Original pfu and APS data available upon request.

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
