# Peer review of "Characterisation of Particle Size and Viability of SARS-CoV-2 Aerosols from a Range of Nebuliser Types Using a Novel Sampling Technique"

_viruses, 2022, doi:10.3390/v14030639_

Round 1

Reviewer 1 Report

This study was designed to determine SARS-CoV-2 viability following aerosolisation, to understand and inform the design of effective animal models utilising an aerosol infection route.

I find this study interesting and relevant. Methods are well described and presented. Conclusions and analysis of the results at a good level.

Comments

1. The authors indicate that relative humidity had no significant effect on viability, size, or number of viable particles produced in experiments. 

I think that it is necessary to specify the time during which the generated droplets were exposed to various conditions (humidity) in experiments and indicate it in the abstract too. After reading the article, it is obvious that this is a short time, which is enough to deliver the aerosol to the respiratory tract of animals - but at first glance after reading the abstract, it may give a misleading impression that humidity does not affect the survival of the virus in the droplets - which is not true (in case of longer exposure).

———————————————-

2. I would like to additionally see the number size distribution of generated droplets/particles (if it is possible).

Based on my experience with various nozzles, atomizers and nebulisers - mass distribution is not always informative in the case of polydisperse atomization of liquids. 

Moreover, I believe that the influence of air humidity will be noticeable precisely in the number size distribution (in some cases the impact can be significant - I observed this in my experiments).

—————————————————

3. I would also like to see (if it is possible) - the presence of fine particles less than 1 micron - and what is the proportion of these particles/droplets in the spray of the atomizers/nebulisers (the number distribution). 

This information can be important for setting up experiments with animals - since it is necessary to simulate experiments with submicron droplets (less than 1 μm) for the spread of viruses (such droplets are generated during normal breathing).

========================

Remarks

I see confusion with punctuation in sentences and references to literature - possibly due to PDF-conversion

line 31 — 

line 34 —

line 42— 

line 54 — 

and so on throughout the text.

Author Response

Reviewer 1:

This study was designed to determine SARS-CoV-2 viability following aerosolisation, to understand and inform the design of effective animal models utilising an aerosol infection route.

I find this study interesting and relevant. Methods are well described and presented. Conclusions and analysis of the results at a good level.

RESPONSE: Thank you for this, always very nice to receive positive feedback for peers

Comments

  1. The authors indicate that relative humidity had no significant effect on viability, size, or number of viable particles produced in experiments.

I think that it is necessary to specify the time during which the generated droplets were exposed to various conditions (humidity) in experiments and indicate it in the abstract too. After reading the article, it is obvious that this is a short time, which is enough to deliver the aerosol to the respiratory tract of animals - but at first glance after reading the abstract, it may give a misleading impression that humidity does not affect the survival of the virus in the droplets - which is not true (in case of longer exposure).

RESPONSE: We thank the reviewer for their insightful responses. A fair comment. The time taken for a particle to travel from the nebuliser nozzle to the sampling points is in the order of 40 microseconds (4.0 x 10-5 sec). This is calculated from: The distance from the nebuliser outlet to the APS/RH probe/AGI (0.7m/0.85m/1m, respectively); the total system air flow (40 lpm) and the spray tube of diameter 5 cm. The spray tube is designed, not to “condition” the generated aerosols, but to ensure even mixing of the generated aerosol, before delivery. In previous work (by ourselves and others, with SARS-CoV-2, and other respiratory viruses such as Influenza) employing the Goldberg Drum apparatus, aerosol particles are held for minutes/hours in a static system, the important influence of RH is observed.

For a particle to be affected by RH within this system it would need to be below 0.5um (see review paper by Božič and Kanduč, “Relative humidity in droplet and airborne transmission of disease”, Journal of Biological Physics (2021) 47:1–29, for some excellent workings of evaporation times); smaller than the APS can measure, and runs a higher probability that these particles are not containing any virus. (see below)

See revised version, text added to both abstract and discussion describing short flight time/RH influence.

———————————————

  1. I would like to additionally see the number size distribution of generated droplets/particles (if it is possible).

RESPONSE: To clarify, reviewer 1 is asking for GSD calculations from CMAD, rather than MMAD? The GSD is automatically calculated from MMAD by the APS device software, and I am unable to provide the distribution of particles from the count values, my apologies. I can share example counts and relative mass data (for each size bin 0.542 to 19.8 um) from our experiments, however, to extract all the raw data from the APS for all 37 runs cannot be achieved within the time frame for us to respond, sorry.

Based on my experience with various nozzles, atomizers and nebulisers - mass distribution is not always informative in the case of polydisperse atomization of liquids.

RESPONSE: in dealing with biological particles, containing microorganisms, one is mindful that the smaller a particle is, the less likely it is to contain a viable microorganism, (rather than just cellular debris, proteins/salts from culture media). As SARS-CoV-2 is circa 100nm/0.1um, particles approaching 0.1um have very low probability of containing what we’re interested in here. For this reason, we have looked to the mass distribution, which tends to give a larger size estimate, to characterise these particles. GSD (indicating the polydispersity) of the CMAD was not calculated by the APS and thus cannot be commented on for this study.

CMAD can be found by applying the formula by Finlay and Darquenne (“Particle Size Distributions”, Journal of Aerosol Medicine and Pulmonary Drug Delivery, Volume 33, Number 4, 2020):

MMAD=CMAD exp[3(lnGSD)2]

Moreover, I believe that the influence of air humidity will be noticeable precisely in the number size distribution (in some cases the impact can be significant - I observed this in my experiments).

RESPONSE: due to the short time frames the aerosols are in the spray tube, RH will only have impacted particles below 0.5um (again, see review paper mentioned in my first response, for these calculations); which are below the detection limit of the APS, and may not contain any virion particles anyhow.

—————————————————

  1. I would also like to see (if it is possible) - the presence of fine particles less than 1 micron - and what is the proportion of these particles/droplets in the spray of the atomizers/nebulisers (the number distribution).

This information can be important for setting up experiments with animals - since it is necessary to simulate experiments with submicron droplets (less than 1 μm) for the spread of viruses (such droplets are generated during normal breathing).

RESPONSE: The authors agree that there is great difficulty in accurately studying microbial aerosols less than 1um in size, but these difficulties are inherent in the science and not something this paper intended to overcome. We feel that the inclusion of the Andersen data helps demonstrates that the majority of particles produced are in the respirable range (fig 3 demonstrates that circa 15% are <1um). Also, given that ACE-2 receptors for SARS-Cov-2 have been demonstrated to be expressed by cells lining the entirety of the respiratory tract, penetration into the alveolar space may not be the only important consideration in causing infection via the aerosol route.

In addition to my previous responses regarding smaller particle sizes, I have made comment in the discussion regarding CMAD vs MMAD.

========================

Remarks

I see confusion with punctuation in sentences and references to literature - possibly due to PDF-conversion: line 31, line 34, line 42, line 54,  and so on throughout the text.

RESPONSE: Thank you for spotting this, duly corrected in revised manuscript

Reviewer 2 Report

Figure 1: APS and diluter would need air flow in and out. And impactor or AGI would need exhaust as well. Please describe those parts.

4 Page, Line 146: Please describe the temperature during experiments.

4 Page, Line 164: Is there any specific reason for dilution? Dilution could lower the precision of measured values during the process due to the loss.

4 Page, Line 171: Please describe cMEM here rather than later description.

Table 2: There seems not to be any relative humidity difference on MMAD. Maybe relative humidity has effects on the small aerosol under 0.5 um. How about number concentration over the sizes which is measured by APS?

7 Page, Line 249: The temperature and relative humidity were kept constant over the experimental conditions?

7 Page, Line 266: Please describe spray factor differences between nebulizers.

8 Page, Line 285: There is an error description. Please edit this.

Author Response

Reviewer 2:

Figure 1: APS and diluter would need air flow in and out. And impactor or AGI would need exhaust as well. Please describe those parts.

RESPONSE: We thank the reviewer for their insightful responses. The image is a representative of the experimental set up; to include all physical connections would make the image too cluttered. I have altered the text describing the figure to reflect this, see revised manuscript.

4 Page, Line 146: Please describe the temperature during experiments.

A good comment, apologies it was not included first time round. See revised manuscript

4 Page, Line 164: Is there any specific reason for dilution? Dilution could lower the precision of measured values during the process due to the loss.

RESPONSE: Dilution is required to bring the number of particles generated into the APS instrument’s reading range: too many particles would saturate the machine. Previous experience from use of challenge solutions of ca. 106 pfu/ml nebulised from the Collisons has shown us that 1:100 dilution brings the numbers of particles generated to within the APS’ limits of detection, allowing for accurate reading of the high numbers of particles generated. All subsequent calculations from the APS take this 1:100 dilution into account.

4 Page, Line 171: Please describe cMEM here rather than later description.

See revised manuscript

Table 2: There seems not to be any relative humidity difference on MMAD. Maybe relative humidity has effects on the small aerosol under 0.5 um. How about number concentration over the sizes which is measured by APS?

RESPONSE: APS cannot measure less than 0.5um, so unfortunately there’s no way of answering that question. Also, in dealing with biological particles, containing microorganisms, one must be mindful that the smaller a particle is, the less likely it is to contain a viable microorganism, rather than just cellular debris, proteins/salts from culture media. As SARS-CoV-2 is circa 100nm/0.1um, particles approaching 0.1um have very low probability of containing what we’re interested in here. For this reason, we have focussed on the mass distribution, which tends to give a larger size estimate to characterise the particles; CMAD tends to be skewed by the large number of very small particles (which may not contain virions) produced. I have made comment in the discussion regarding CMAD vs MMAD.

7 Page, Line 249: The temperature and relative humidity were kept constant over the experimental conditions?

RESPONSE: In comparing the 6 different nebulisers, temperature was constant, RH was changed for the nebuliser study, to fall within 3 ranges: <45%, 45 to 60% and >60%. In evaluating the Andersen sampler, RH and Temp were kept constant. Apologies this was not clear, text at line 176 page 5 was altered to include this clarification.

7 Page, Line 266: Please describe spray factor differences between nebulizers.

See revised manuscript

8 Page, Line 285: There is an error description. Please edit this.

Apologies, see revised manuscript